# The Bible as a Successful Migrant? Translation, Domestication, and Nordic National Identity

Karin Neutel

Department of Historical, Philosophical, and Religious Studies, Umeå University, 907 36 Umeå, Sweden; karin.neutel@umu.se

**Abstract:** Despite its ancient and foreign origin, the Bible has managed to integrate so seamlessly into the contemporary Nordic countries that it is seen to form the basis of specifically Danish, Norwegian, and Swedish identity and values. This paper will employ the work of translation scholar Lawrence Venuti on the politics and ethics of translation, and especially his concepts of foreignization and domestication, to explore this understanding of the Bible. Venuti's thought informs critical reflection on how translation contributes to the cultural position of the Bible in the Nordic context, through examining translation principles, the function of retranslation, and the role of fluency. This contributes to our understanding of how the Bible has become a successful migrant to the Nordic region who is now used to keep others out.

**Keywords:** Bible; translation; Nordic countries; Lawrence Venuti; politics

## 1. Introduction

When thinking about the Bible and migration, the first thing that comes to mind might be stories of migration and travel in the biblical canon, or the role that biblical stories can play in the lives of people moving in today's world. While other articles in this special issue will explore both these aspects, this paper approaches the topic of the Bible and migration from a different angle: it takes the Bible itself as a migrant. I suggest here that we can see the Bible as a very 'successful' migrant to the Nordic countries Denmark, Norway, and Sweden, where the Bible is described as a book that supports 'our' values, and that is key to understanding and transmitting Nordic national identity.[1] The Bible appears as a 'successful migrant' by barely showing any signs of its ancient foreign origins and instead aligning with what is seen as important for being Danish, Norwegian, or Swedish. This sense of the Bible being 'ours', as supporting what 'we' value and how 'we' live, thus also allows it to also be used in support of anti-migration and nationalist politics.

The understanding of the Bible that is the focus here erases the geographical, cultural, and temporal distance that separates the biblical source texts from a contemporary Nordic audience. This erasure allows these texts to be relevant in constructing a present-day Nordic national 'us'. The point of applying the metaphor of a 'successful migrant' to this type of understanding is not to deny a connection between biblical texts and Nordic culture, but rather to draw critical attention to its framing. The close link between the Bible and Nordic national identity not only makes its Middle Eastern origins invisible, but also ignores that the Bible can be claimed to have a similar significance for people in Colombia, Kenya, and the Philippines, for example, as it has in Denmark, Norway and Sweden.[2] By portraying the Bible as being at home especially in these Nordic countries, the Bible's belonging is limited, as is its role in creating belonging for its historical and current global audiences.

This article focuses on the role of translation in this national positioning of the Bible. For my analysis, I use the work of translation scholar Lawrence Venuti, who examines ethical aspects of translation trough the concepts of 'foreignizing' and 'domesticating' (I will use the pairs 'foreignizing' and 'foreignization' and 'domesticating' and 'domestication' interchangeably and without quotation marks from here on). Venuti's work is particularly helpful to clarify how the Bible functions in Nordic national contexts, which tends to be in its translated form.

## 2. The Bible and Nordic Identity

The integration of the Bible into Nordic society—its status as what I describe as a 'successful migrant'—can be broken down into different aspects. In this section, I highlight three aspects that I see as especially significant based on my own research, although other aspects could certainly also be included.[3] First, the national status of the Bible is affirmed quite directly by national Bible societies, particularly when new Nordic Bible translations are produced and marketed. The Bible is explicitly described as being an important source of national values and fundamental stories and as connected to a specific national 'us'. Second, the status of the Bible is also evident in its use in politics, particularly as a source that supports not a specifically religious or Christian view, but rather a broader view of society and politics that is often assumed to be widely shared within a national context. A final aspect is the connection between the Bible and Nordic royalty, which shapes the position and status of the Bible in perhaps a less direct but still forceful way. While this integrated, national positioning is not the only way the Bible is understood, and it is not understood this way by everyone, I argue that it is important to recognize this understanding as a feature of the Bible in the Nordic context, especially in relation to the topic of migration.

### 2.1. Nordic Bible Production

The first aspect of 'successful' integration of the Bible is presented by those who produce Bibles in Nordic countries, especially the national Bible Societies of Denmark, Norway, and Sweden. Some examples from each of the three countries will illustrate the pattern, particularly in and through their marketing efforts.[4] When the Norwegian Bible Society celebrated its 200-year anniversary in 2016, it did so with slogans pointing to the Bible's national significance. Buses drove through Oslo with the motto 'The Bible for everyone' on the side, and the Bible Society commissioned a publication titled 'The Book that Shaped Our Culture', since 'no book has influenced our culture as much as the Bible' (https://bibel.no/nettbutikk/bibelen-boken-som-formet-v%C3%A5r-kultur (accessed on 7 March 2025)).

The introduction of the most recent official Swedish Bible translation, *Bibel 2000*, similarly connects this text to a national collective: 'It is by being put into general use by the Swedish people that the translation achieves its stated purpose: to be the main text for the Bible within the Swedish language area.' This new translation is not intended for Swedish Christians or the Church of Sweden, but for the Swedish people as a language community. This is partly a result of the particular origin story of this translation, a translation initiated and funded by the Swedish government before the separation of church and state, finalized in the year 2000.[5]

The rhetoric used in Denmark can be even more emphatic. The Danish Bible Society announced its *Bibelen 2020* translation into Nudansk and the official translation planned for 2036 with the announcements 'Denmark gets a new Bible', or even 'All of Denmark gets a new Bible' (https://www.bibelselskabet.dk/hele-danmark-faar-en-ny-bibel; https:

//www.bibelselskabet.dk/bibelen2020 (accessed on 7 March 2025)). In their press material for *Bibelen 2020*, the Danish Bible Society writes:

> The biblical stories are essential stories in our culture. They should of course be accessible, so they can be read and understood by modern people and experienced as still applicable and relevant. In Bibelen 2020, God speaks modern Danish. Is this possible for God? Of course. God speaks the language that we speak (Pressemeddelelse: "Danmark Får Ny Bibel," Bibelselskabet, 2020).

The Bible is presented as a resource that needs to be accessible because it is essential to Danish culture. Bible translation is a means by which God can sound like a contemporary Dane, because God speaks whatever language 'we' speak. This type of legitimation also forms the basis for the new official translation which was recently initiated. The chairman of the board of the Danish Bible Society, Peder Ø. Andreasen, explains the fundamental significance of this translation:

> Therefore, there is a need again and again to translate the Bible and strengthen knowledge of the stories that bind Danish society together. If we lose them, we lose ourselves. . .A new Bible translation should be for joy, for faith, for reflection, for community—for us as a country and people (https://www.bibelselskabet. dk/danmark-skal-have-en-ny-bibel (accessed on 7 March 2025)).

In these examples, Norwegian culture and a Norwegian sense of 'everyone', the Swedish language and people, and Danish culture and society are rhetorically connected to the Bible as a significant, or even *the* most significant source and text. Of course, in all three countries, there are people who, for religious, ethnic, linguistic, or other reasons, might object to this appeal to a singular national language and culture. Official languages in the countries discussed here include not only Danish, Swedish, and the Norwegian languages Bokmål and Nynorsk, but also Sami, Kven, and Meänkieli. Given the contested status of minority groups and languages in the Nordic region, the types of identifications discussed above have significant socio-political as well as specifically inclusionary and exclusionary implications.[6]

The fact that these Bible Societies actually produce Bibles in multiple languages in order to serve diverse audiences in these countries is an indication of the complexity and plurality that is belied by the rhetoric on national identity cited above. The 'Swedish people' mentioned in the introduction of *Bibel 2000* might read the Bible in Northern Sami, in English, in another Swedish version, or not at all, rather than have an obvious connection as a people to this one text. The producers of Bible translations thus draw from and contribute to this sense of the Bible being important for a national 'us'. They promote and legitimize their translations by framing the Bible as having a unique connection to specific linguistic majorities in Nordic countries. Section 4 below will further explore the role of Nordic translation in this national framing.

### 2.2. The Bible in Nordic Politics

While Nordic countries have a reputation for being largely secular, the Bible plays a larger role in political discourse than is often assumed. Much work still needs to be done on mapping this role, yet recent research on the reception of the Bible in Nordic contexts shows that, for all three countries, the Bible appears as a source of legitimation in political discourse that appeals to a wider socio-cultural, rather than a more narrowly Christian understanding.[7] This can look different in terms of the texts that are referenced, but there appear to be significant similarities in the use of the Bible as a political resource across all three countries.

In previous research, I have shown how the concept of neighbourly love and the story of the Good Samaritan (Luke 10:25–37) are used in Denmark, Norway, and Sweden, as

well as in Germany and the Netherlands, to support a politics that is critical of or hostile to migration (Neutel 2022; Neutel and Bjelland Kartzow 2021). While some politicians connect neighbourly love explicitly to Christian values and identity, they tend to enlist this concept for a national interest that is conceived more broadly, rather than for a specific religious group. While 'neighbourly love' is thus used politically across Nordic countries, there are also biblical references that are more specific to one country. Kaper Bro Larsen has studied the wide use of the biblical text Mark 12:17 in Denmark: 'Give to Caesar the things that are Caesar's and to God the things that are God's' (Larsen 2023). This verse was cited in 415 articles in Danish national newspapers since 1 January 2000, although it was not always identified as a biblical reference, occasionally attributed to Martin Luther.

The phrase was central in the response by the then Prime Minister of Denmark, Anders Fogh Rasmussen of the Liberal Party Venstre, to the so-called 'cartoon crisis' around the depictions of Mohammed in a Danish newspaper in 2005–2006. In an opinion article for a national newspaper, Fogh Rasmussen claims to be 'heavily influenced by Jesus' famous words' about giving to Caesar (Larsen 2023, p. 83). He argues, rather paradoxically, based on this saying that there should be a clear distinction between religion and politics, and that religion should be kept indoors. The verse is referenced predominantly by center-right and right-wing nationalist parties, in what Larsen calls 'Lutheran secularism discourse' (Larsen 2023, p. 95). Larsen distinguishes five main discourses by which the saying serves to define the limits of (1) religion, (2) the state, (3) nationhood, (4) Islam, and (5) religious free speech rights. When used by contemporary Danish politicians, Jesus' saying becomes a proof text that legitimizes their interpretation of these five discourses while simultaneously promoting an idea of the Bible as a foundation for modern secularism and rights.

This particular reference appears to be prevalent in Denmark and is not used similarly in Norway and Sweden. However, cases from these countries confirm that the types of purposes and implications biblical references tend to have in a Nordic context are largely similar. In analysing references to the figure of the 'Mammon' (Matthew 6:24; Luke 16:9–13) and to Jesus' expelling merchants and money changers from the Temple (Mark 11:15–18, Luke 19:45–47, John 2:14–16), that were made in recent Swedish debates on health care, Hanna Stenström describes what occurs as 'receptions of receptions'. These references constitute an appeal to shared cultural heritage used to support political positions that are not explicitly identified as Christian (Stenström 2023, p. 116). Ole Jakob Løland notes a similar cultural use of the Bible in Norway. He observes a hesitance on the part of Christian Democratic politicians to claim biblical legitimacy and a willingness by the right-wing Progress Party to position themselves instead as connected to biblical and Christian legitimacy, particularly around migration and Islam (Løland 2023).

Biblical references thus tend to take a different form than the 'cite and run' type of references that may be familiar from political Bible use in the United States and other international contexts (Berlinerblau 2007). The purpose of the reference is not to claim a specific religious authority or legitimacy by using scriptural proof texts, but rather to appeal to a widely shared cultural resource. It seems likely that this understanding of the Bible as a cultural resource is connected to and builds on the type of national identification created by Bible producers, as discussed above.

### 2.3. The Bible's Royal Connections

Another aspect of the role of the Bible in Nordic identity that rarely receives attention, and that I therefore want to highlight here, is its connection to royalty. Bible production has long depended upon those in state power, from Emperor Constantine in the fourth century, when the very first copies were produced, but little scholarship on this link exists

(see Section 4.1 for more). For the Nordic countries considered here, this link is relevant for the public, political, and national profile of the Bible.

Whenever there are festivities around the Bible, when a new translation needs to be presented or an anniversary is celebrated, members of the royal families are involved. Conversely, when royals want to express their public role, they can do so by using the Bible. For example, Swedish King Carl Gustav and Queen Silvia were presented with the Northern Sami Bible translation by the Swedish Bible Society in 2020 in a public ceremony, as has happened with other Bible translations. The Swedish royal family also has biblical content on its own YouTube channel. In a 2023 Christmas video, Crown Princess Victoria appears reading the Nativity story, in order, as she explains in her introduction, 'to come together for a moment and remember why we celebrate this holiday'. During the 200-year anniversary of the Norwegian Bible Society mentioned above, the Norwegian Crown Princess Mette-Marit read verses from Psalm 71 as part of a Bible marathon, where the entire Bible was read aloud in a park in Oslo. The 200-year anniversary of the Danish Bible Society was celebrated in 2014 with a church service attended by 'Her Majesty Queen Margrethe as well as prominent politicians and artists'.[8]

In all three countries, the current royals have roles as patrons of the national Bible societies. The new Danish King Fredrik, for example, took over from his mother, Queen Margarethe, as patron of the Danish Bible Society. In Denmark, which has a national or 'folk' church, the reigning king or queen authorizes the official Bible translation. The former Danish queen not only authorized the 1992 Bible, but also provided illustrations for editions of the Bible in Danish, Greenlandic, and Faroese. Her involvement with the Danish Bible Society is honoured on the society's website with a photo series, titled '52 years with the Queen and the Bible' (https://bibelselskabet.dk/billedserie-52-aar-med-dronningen-og-bibelen (accessed on 7 March 2025)) which cites her motto: 'God's help, the people's love, Denmark's strength'. As all of these examples show, the sense of the Bible as a national text is strengthened by its very public connection with Nordic monarchies, who in turn can use the Bible in their role as representing a national 'we'.

### 2.4. A Brief Comparison with Other National Bibles

To place these Nordic understandings of the Bible as a national text in a more international perspective, it is helpful to compare with other countries. Given limited space, I focus on examples from the United Kingdom and the United States.

Like his Nordic counterparts, King Charles is also the patron of a bible society, the British and Foreign Bible Society (BFBS). The ceremonial visibility of the Bible through its British royal connections extends not only to a national but also to a global scale. The BFBS pointed out in May 2023 that the funeral of Queen Elizabeth II 'was the largest public Scripture event in history, with millions of people around the world hearing words of Scripture said and sung'. The same article on the Society's website notes that 'the coronation of King Charles III may well be the second largest'. During the coronation ceremony, the King was presented with a Bible as a gift, which he then kissed (Lloyd 2023). The notes to the coronation liturgy explain the significance of this gift as 'a reminder that careful and prayerful attention to the Bible is at the heart of Christian worship and devotion, as well as being the historical foundation of so much of Britain's culture and ethics'.[9] The Bible that the new King was presented with was one of which only four copies were produced, with King Charles' cipher embossed on the spine of a red leather cover and containing 'the full King James Bible, in traditional typography' (Artman 2023).

It is probably not surprising that when the Bible appears in a British ceremonial setting, it is in the form of the 17th century King James translation, rather than in a contemporary

version. Much has been written about the cultural significance of this particular text, especially on its 400th anniversary in 2011.[10] However, compared with Nordic national Bible use, its appearance is an important observation and noteworthy divergence. When a new Bible is produced for a historic British royal occasion, it is a reiteration of a classical, rather than a modern text. This attachment to the cultural significance of the King James Bible was also evident in the decision by then Education Secretary Michael Grove to mark the 400th anniversary by donating a new copy of the King James to every state school in England, which included a brief foreword written by Grove himself. Grove is quoted as calling the King James Bible 'one of the keystones of our shared culture' and describing the translation of the Bible into the vernacular as 'a critical moment in the life of the nation' (https://www.christian.org.uk/news/schools-set-to-receive-a-king-james-bible-from-government/ (accessed on 3 March 2025)). While this vernacular translation is repeatedly redone in a Nordic context to maintain its national function, one of the earliest translations into English can continue to function with national significance in a British context.

The recent *God Bless the USA Bible*, launched in 2024, provides a helpful illustration of an attempt to produce and market a new national Bible in the US.[11] It has several features that express its national identity. On the cover, in addition to the words 'Holy Bible', the name 'God bless the USA' appears over a depiction of a waving US flag. Inside, the Bible includes the 'trusted King James translation', the US Constitution, the Bill of Rights, the Declaration of Independence, the Pledge of Allegiance, and part of the lyrics to the song 'God Bless the USA'. It is especially the inclusion of supplementary texts and imagery with national political and cultural significance gives this Bible its connection to a particular national context. The translation alone would likely not be sufficient to create such a national identification. This example shows that the type of additional material required to make this Bible believably 'American' is not necessary in the case of creating a Nordic national identity. Nordic national languages provide a crucial national connection—although not in uncomplicated ways, as addressed above—and the exterior of Bibles and their naming do not need to contain national symbols.

Having recognized some key features of Nordic national Bibles, understood in comparison with international examples, we can now turn to Lawrence Venuti's work and the possible role of translation in shaping how Bibles function.

## 3. Venuti and the Ethics of Translation

Lawrence Venuti has been an important voice in Translation Studies since the 1990s, particularly in emphasizing the significance of cultural contexts for translation practices, as well as in encouraging translators to consider the broader ethical, social, and political implications of their choices (Venuti 1995, 1998, 2013, 2021). Venuti notes something of a paradox in the translation process, particularly in connection with national contexts: translation introduces difference into a national context and can be seen by some as an act of violence against this context by introducing something foreign. Yet, the status of a language and culture is simultaneously presupposed as well as created and affirmed through translation: 'Translation can support the formation of national identities through both the selection of source texts and the development of discursive strategies to translate them' (Venuti 2013, p. 119).

Central to Venuti's work is the idea of translation as a practice that mediates between cultures. This mediation can be done in different ways, depending on the strategies translators use to handle cultural differences between the source and target languages. Venuti distinguishes between two dominant strategies: foreignizing and domesticating. Foreignizing refers to foregrounding foreignness and difference in the translated text.

Foreignization retains elements of the source text's culture, thereby deliberately breaking the conventions of the target language to highlight the text's foreign origins. It introduces perceptible differences within the target culture, making the reader aware of the translation process and cultural differences between source and target. Domesticating, on the other hand, makes the translated text closely conform to the cultural norms of the target language. This approach aims to make the text more accessible and familiar to the target audience. Domestication refers to translations aiming for fluency, naturalness, and easy readability. Fluency is an important characteristic that Venuti associates with the invisibility of the translator. Fluency means familiarity; fluency enables the target society to ignore and suppress differences with the source. The use of standard dialect as well as clear syntax and meaning 'creates an easy readability that masks the translator's work, leading the reader to believe that the translation is actually the source text' (Venuti 1995, p. iix). In considering the ethical implications of translation in connection with foreignizing and domesticating, Venuti emphasizes the risks of domestication:

> For "domesticating" and "foreignizing" are ethical effects whereby translation establishes a performative relation both to the source text and to the receiving situation. Domesticating translation not only validates dominant resources and ideologies, but also extends their dominance over a text written in a different language and culture, assimilating its differences to receiving materials. Thus domesticating translation maintains the status quo, reaffirming linguistic standards, literary canons, and authoritative interpretations, fostering among readers who esteem such resources and ideologies a cultural narcissism that is sheer self-satisfaction (Venuti 1995, p. xiv).

Interestingly, Venuti points out how *re*translation is likely to amplify domesticating effects. Retranslation can be seen to negotiate with the target context in two ways, firstly through the general translation process and then again in relating in some way to one or more previous translations that are specific products of the target context. The outcome of this is likely to be 'doubly domestic':

> Translation is an inscription of the foreign text with intelligibilities and interests that are fundamentally domestic, even when the translator maintains a strict semantic equivalence with the foreign text and incorporates aspects of the foreign-language cultural context where that text first emerged. Retranslations constitute a special case because the values they create are likely to be doubly domestic, determined not only by the domestic values which the translator inscribes in the foreign text, but also by the values inscribed in a previous version (Venuti 2004, p. 25).

A further significant dimension of retranslation identified by Venuti is that of strengthening the authority of the social institution responsible for a retranslation by reaffirming the institutionalized interpretation of a specific text, especially one that is canonical. This effect can be intensified when a retranslation intends to challenge a previous one. Here, Venuti points to the King James translation in relation to the authority of the Anglican Church during the early seventeenth century (Venuti 2004, p. 26). Even more than regular translations, conscious retranslations highlight the intentionality of the translator, since a retranslator aims to bring about a new and different reception for a text already present in the translating culture. In challenging a previous version of the foreign text, a retranslator constructs 'a more dense and complex intertextuality' and therefore ties the text even more closely to the target culture (Venuti 2004, p. 32).

Because of his concerns about the ethical consequences of domestication, Venuti mainly advocates in favour of foreignizing translation, arguing that it respects the otherness of

the source text and its cultural uniqueness, allowing readers to experience the original culture more authentically. As a translational practice, foreignization also has the potential to challenge the dominance of the target culture by introducing and valuing cultural differences.

The precise cultural and ethical effects of any translation, of course, depend on the contexts of the source and target languages. Since Venuti works mainly on translations into English, he associates the target language with cultural dominance, but also acknowledges that processes of translation can have very different effects depending on the circumstances: 'it would be reductive to attempt any ethical or political evaluation of translation nationalisms without considering the historical moments in which they emerged' (Venuti 2013, p. 140).

While Venuti's concepts of foreignizing and domesticating have been welcomed, they have also sparked debate within translation studies. A general issue is the fuzziness of the concepts and the presumed boundaries between the two approaches, which can be seen as an oversimplification, since not all translations may fall into these binary categories (Myskja 2013).

More problematic for the case we are discussing here is that Venuti's own category of the foreign has exoticizing tendencies, which may reinforce cultural stereotypes and reify cultural otherness. As Tarek Shamma points out, foreignizing translation is a tool that, in principle, could serve many different ideological agendas (Shamma 2009). In addition, the specific situation of minority languages raises concerns for the applicability of Venuti's ideas. As noted, foreignization will have different consequences when translating into a minority language threatened by other dominant ones and might exacerbate the marginalization of such a language (Myskja 2013). In discussing the complexities of translation ethics, Maria Tymoczko notes that Venuti's promotion of foreignization misses some contextuality:

> Foreignization may be appropriate for dominant cultures such as the United States, but it is not suited to subaltern cultures that are already flooded with foreign materials and foreign language impositions. Foreignization has also been rightly criticized as a potentially elitist strategy, more appropriate to a highly educated audience than a broad readership (Tymoczko 2006, p. 454).

Considering these critiques about how foreignization and domestication may function in relation to majority and minority cultures and languages, we now explore the applicability of Venuti's concepts and their ethical implications for translating the Bible in Nordic national contexts.

## 4. Venuti and the Bible in Nordic National Contexts

Venuti's ideas about foreignization and domestication, and the role of fluency in hiding the translator's in/visibility, have great explanatory potential when applied to Bible translation in the Nordic context. Important to note is that, in developing these concepts, Venuti does not engage with contemporary Bible translation, nor with the Nordic contexts that this article focuses on. He does relate to the work of Eugene Nida and historical Bible translation—as is almost inevitable in the field since the Bible is the most frequently and widely translated text.[12] Venuti's own analyses are mainly based on the translation of literature and poetry into English, often from Italian, French, and Catalan.

Venuti's work not only helps to understand the process of translation considered here, but also clarifies the effects that this type of translation is likely to produce. Several of the effects of domestication as described by Venuti appear strikingly applicable to recent Nordic Bible translations and their connection to national identity. However, Venuti's concepts may also relate to these translations paradoxically, such as when it comes to the invisibility of translators. In this section, I consider several aspects of the Nordic Bible as a domestic Bible and the role of translation. I begin with the impact of Nordic translation

principles and practices more generally. The second aspect relates to the significance of retranslation. Finally, there is the role of fluency, which is not only an explicit aim when it comes to the written text, but also extends to the translation when read aloud.

### 4.1. Nordic Translation Principles and Practices

Bible translation has a long history in the Nordic region, with partial translations in Old Norwegian possibly appearing as early as 1300. The first full Danish and Swedish New Testaments were published in 1524 and 1526, respectively. While these translations were inspired by and quickly followed Luther's 1522 translation, they actually preceded the Reformation in the Nordic region by quite some time and 'served to prepare the soil for it in Scandinavia' (Noack 1963). In this region, as elsewhere in Europe, early Bible translations were initiated by and named after rulers (e.g., Gustav Vasa, Christian III, and Frederick II). From the beginning, these vernacular translations could be seen as authoritative texts in their own right. In the preface to his Danish translation of the Book of Homilies, which also included the New Testament Gospels and letters, one of the earliest Bible translators, Christiern Pedersen states that: 'Nobody ought to think that the Gospels are more sacred in one tongue than in another: they are as good in Danish or in German as they are in Latin, if only they are rightly interpreted' (cited in Noack 1963, p. 137).

The idea that a translated Bible functions as an authoritative form of the text on a par with other forms is thus not only the byproduct of a type of translation in the Nordic context, but a matter of explicit ideology. Bible translations have been able to function here as equivalents of the source text, even if the status of particular translations has also at times been a matter of dispute. The illusion where the translation becomes the source text for the reader, which Venuti associates with domesticating translation, is thus an explicit feature of the Bible in its translated form for Nordic readers and has been so for many centuries. This is likely also to inform how translation is done. It is easy to hear an echo of Pedersen in the statement cited above from the Danish Bible Society that 'God speaks our language', which both explains and justifies a translation into contemporary Danish.

The Danish translation, *Bibelen 2020*, has explicit translation principles that prioritised the target culture, aiming for 'flow and understanding', and avoiding words that may sound archaic to contemporary Danish ears (Neutel and Bjelland Kartzow 2023). In a review of *Bibelen 2020*, Gitte Buch-Hansen characterizes this translation into Nudansk by citing Friedrich Schleiermacher's notion that the translator leaves the reader in peace and moves the author towards the reader (Buch-Hansen 2020). This characterization neatly identifies it as a domesticating translation as defined by Venuti, where the reader is presented with a text that seems familiar and does not challenge them. Based on analysis of translations of the fifth commandment in Exodus 20 into Nordic languages, Søren Lorenzen concludes that the rendering of one commandment—'Respect (Du skal respektere) your father and mother', rather than the earlier Danish translation 'Honor (Ær) your father and mother'—'clarifies more about contemporary society than an ancient one' (Lorenzen 2023, p. 159).

For the new official Danish Bible translation planned for release in 2036, with the working title *DO36*, the Danish Bible Society is using somewhat different translation principles, which appear more focused on tradition. In a publication on these principles, the context for *DO36* is described as follows:

> DO36 writes itself into a century-long tradition of text-focused translations that are authorized for use in the Danish Church, that are widely used in general church life, and have a central place in homes, schools, and culture. At the same time, a translation that is to be published in 2036 and to remain relevant for decades must dare to go in new directions and look beyond the familiar. DO36 is thus intended as a translation that will meet the tension between

the familiar and the innovative (Bibelselskabet, Principper for ny autoriseret oversættelse af Bibelen).

*DO36* is positioned as a retranslation that is aware of its own relationship with previous translations. The Danish Bible Society formulates four principles for this translation, which are imagined as four corners of a trampoline. These four principles are the 'source text', the 'Danish language', 'tradition', and 'use'.

The first principle of the 'source text' is where biblical, philological, and historical scholarship comes in, and thus, the greatest potential for foreignization through recognition of the source context. This is formulated here quite explicitly in that the translation 'allows foreign elements in the source text to be expressed'. Yet there is a tension here with the second principle of the 'Danish language', which is taken to mean that the translation must 'strive for an idiomatic, understandable, and natural language that is not already outdated in terms of vocabulary, grammar, and spelling'.

The other two principles, 'tradition' and 'use', also seem to stand in tension with the possible foreignization allowed under the first principle, and tend more towards domestication. In discussing 'tradition', the society emphasizes that translation should be aware of key theological and liturgical terms, and of 'wording with a strong anchoring'. For 'use', this retranslation should be able to be used broadly, by and for everyone, in Christian as well as broader cultural and societal contexts. They state, 'therefore, the translation must, to the greatest extent possible, be colloquial and suitable for reading in order to make sense to those who hear it read aloud'.

These four principles are imagined by the Bible Society to form the four corners of a trampoline: 'In other words, the four considerations can stretch out a trampoline that provides energy for development'. Given the imbalance between the four corners when understood in terms of foreignizing and domesticating tendencies however, it seems likely that the trampoline will end up rather saggy, since the domesticating corners have the greatest pull. This might not provide much energy to those elements that have the potential to challenge cultural and linguistic expectations.

While the metaphor of the trampoline appears to be unique to the latest Danish translation, the principles that are formulated here are in many ways similar to the dynamics that inform the most recent Bible translations in Swedish (NT2026, initiated in 2022) and Norwegian (Bibel 2024, a revision of Bibel 2011, published last year). For both the Norwegian and Swedish translation projects, the biblical scholars in charge, Jorunn Økland and Mikael Winninge, have emphasized the need to integrate recent findings from Biblical Studies into translation.[13] Yet the focus on source culture that is a part of this scholarship inevitably stands in tension with other translation goals, as neatly expressed in the Norwegian Bible Society's description of *Bibel 2024*: 'a professionally strong and user-oriented translation', that is the result of contributions by 'our best professionals and literary authors'. As a revision of *Bibel 2011*, *Bibel 2024* has chosen to maintain previous translation principles, but is somewhat less concerned with aligning translation to contemporary intelligibility:

> We have also worked to show the literary style and structures used in the Bible, because through this the translation becomes more faithful to the culture and time in which it was written. The purpose is to approach the Bible as honestly as possible—where it is difficult for today's readers to understand, it must be (https://bibel.no/https-bibel.no-bibel2024 (accessed on 5 March 2025)).

The Swedish *NT2026* follows a similar pattern. It recognizes that most translations operate 'on a continuum between formal and functional equivalence' and expresses a desire for the current translation 'to realistically reproduce foreign and culture-specific features in the text, without adapting them to modern times'. It distinguishes central

terms and concepts, which should be 'reproduced as consistently as possible', from the linguistic structure and imagery, which can be 'adapted to the conditions of the target language' (Winninge 2021). Contemporary Nordic Bible translation is thus caught in inescapable tensions between the potential foreignizing impact of biblical scholarship, and the domesticating aim of producing a text that serves contemporary readers in religious and cultural settings which lives up to the idea of the Bible as a sophisticated literary text.

Maria Tymoczko's critique of foreignization as a potentially elitist strategy suitable especially to an educated audience also seems relevant here. It is a point of pride for Nordic Bible Societies that each new Bible translation is produced by academic experts and contains the latest scholarly insights. The Bible is a text that changes and becomes even better with each new version, while at the same time maintaining its foundational cultural role, partly through its updatability. Even in its foreignization, then, these translations cannot escape domesticating effects.

### 4.2. Bible Translation as Retranslation

An important aspect of this updatability is the fact that Bible translation is a conscious *re*translation. As the principles outlined above show, each new translation explicitly places itself in a national and sometimes international tradition of translation. Translating the entire Bible with experts on specific languages and texts is a costly and long-term endeavour, which makes revisions of previous translations an attractive alternative. In addition, because Bible translations tend to take on cultural and liturgical significance, any changes to familiar wording mean changes from previous translations tend to be the focal points of critique for any new revision or retranslation. Nordic Bible translations are thus likely to be susceptible to the doubly domesticating effects of retranslation as depicted by Venuti. They are determined not only by the domestic values the most recent translator inscribes, but also by the values inscribed in previous versions. It is not primarily the foreign text that forms the impetus for translation, but rather the experience of the last translation no longer meeting current needs or standards.

The effects of retranslation on the authority of the social institution producing the text are also at play in the Nordic contexts, where Bible societies compete with other producers of Bibles in Nordic languages. These other producers, however, often lack the financial, social, and academic resources and connections that national Bible societies have, including their links with the monarchy. In a varied landscape of translated Bibles, the retranslations produced by the Bible societies can therefore raise the profile of these societies as actors that help shape national identity.

A final feature worth highlighting in this context is the names given to recent Nordic Bibles. As is clear from the examples mentioned in this article, these tend to be simply the word 'Bibel' or 'Bibelen' ('Bible' or 'the Bible'), or an abbreviation such as 'NT' followed by the year in which the translation is published. Unlike previous Bibles, which were often named after kings or included references to God or a church, what distinguishes these Bibles is only the year of publication. This naming practice has a technological flavour, as if the Bible of this particular year is simply the most updated and advanced version of a similar previous product. No specific individual identity is conveyed for any translation; instead, the suggestion is that the latest version is the one that will be most compatible with other aspects of current daily life. This also means that any translation is dated in a literal sense, and will seem outdated quickly. Relevant in the context of foreignization and domestication is that if a Bible named in this way starts to seem outdated, it is by being several years or decades old, rather than a number of millennia.



*4.3. Fluency and the Invisibility of the Translator*

As evident from the abovementioned principles, fluency is another critical and constitutive characteristic for Nordic Bible translation. An important impetus for new translation comes when the language of the previous translation appears old-fashioned. The aim is to have a translation that sounds contemporary yet will endure for some time. This emphasis on fluency, one of the four main principles in the new Danish translation and an important consideration across Nordic translations, is likely to have domesticating effects.

These effects are heightened in the case of Nordic translation, in that the aim tends to be to find a text that not only appears fluent when read by individual readers, but also when it is read aloud and heard. As quoted above, the new official Danish translation should 'to the greatest extent possible be colloquial and suitable for reading in order to make sense to those who hear it read aloud'. Similarly, the Swedish *Bibel 2000* translation had the ambition to work in private and public reading 'from the pulpit, on stage and in the studio'. To meet this extended criterion of fluency that includes how the text sounds to people who do not have it in front of them, as well as how easy it is to read aloud, a translation will likely need to be very close to what appears as natural target language.

Venuti connects the domesticating effects of fluency to the invisibility of the translator. Since the text is experienced by readers and, in this case, also listeners as being fluent, it appears as an original contemporary target language text rather than originating from another time and place. This may well be the effect that fluency creates in the relationship between the reader and the translated biblical text, but it is contradicted by the heightened public visibility of the actual people involved in Bible translations, especially at the moment when a new Nordic translation is launched. In the most recent Swedish and Norwegian translations, literary authors such as Karl Ove Knausgård played a well-publicized role, focused on safeguarding the literary expressions and techniques used by the biblical authors. The involvement of these literary figures seems to underline both the literary quality of the Bible as well as its position as a cultural icon. The authors who contributed to the Norwegian *Bibel 2011* translation subsequently published a volume on their experiences that also included new work inspired by biblical texts (Amadou and Aschim 2011).

In light of Venuti's analysis, there is thus something paradoxical about the remarkable degree of visibility achieved by these translators, whose role was explicitly aimed at fluency and a sophisticated target language experience. Awareness of the translation process, the funding and people involved, the research and expertise behind it, which are all publicly presented when a new translation is launched, does not seem to challenge the perception of the Bible as 'our' text, but may rather work to strengthen this perception.

## 5. Concluding Thoughts: The Bible as a Migrant

The ways in which the Bible functions in a Nordic context and the claims about its meaning for national culture and identity are multidimensional and complex. Translation only forms one part of this complexity, but it is a significant one. As the application of Venuti's work here shows, the principles and practices that inform translation in the Nordic context allow the Bible to be experienced like a Nordic contemporary. Through the ongoing process of translation and retranslation, which is aimed at present-day fluency and recognizes tradition and continuity as important aspects, the Bible is thoroughly domesticated. It appears not as a temporal and spatial foreign 'other', not as a text that is identifiable as translated from a different cultural context, but rather as a product of the receiving culture.

This has implications for the Bible's status as a migrant, as well as its use in discussions about other migrants. The exclusionary political rhetoric discussed in Section 2.2 is not an isolated phenomenon, but draws on this wider cultural sense of the Bible being 'ours'. The

explicit boundaries that are drawn in Nordic politics in applying biblical texts to migration build on the boundaries that are implicit in the nationalistic framing of the Bible in its production and royal connections. Translation is especially significant here in creating the illusion that the Bible is more Danish than, for example, Columbian, or is Norwegian rather than Philippine. The Bible could be portrayed as creating connections across time with the Middle East, and across global space today. Instead, the Bible's current Nordic framing suggests that it is uniquely at home here, and that it has significance for everyone who shares this home.

Research on the discursive conceptualization of migration can add to this metaphorical understanding of the Bible as a specific kind of migrant. As Catrin Lundström argues in her study on 'white migration':

> The discursive concept of "the migrant" tends to be used as a marker of non-whiteness and a non-Westerner (...). In this conceptual conflation, non-white bodies "out of place" tend to be (mis)read as being (first, second or even third generation) migrants, "illegal immigrants" or "asylum seekers", despite their possible citizenship in the country in which they reside (...). On the contrary, "white migrants" can inhabit the world as part of a global enterprise, tourists, expatriates, guests, development aid workers, and so on, representing humanity, whose presence remains undisputed or who are able to use their white ethnicity as a form of "symbolic ethnicity" (...). The question of migration is therefore intimately connected to the politics of mobility, its restrictions and possibilities (Lundström 2014, p. 2).

The extreme mobility and translatability of the Bible and its domestication through translation allows it to function in a Nordic context as a 'white migrant' who is not read as being 'out of place' and recognizably foreign. Rather, the Bible is part of a global enterprise, representing humanity, in no way limited by its Middle Eastern origins. At the same time, the Bible is read as recognizably and significantly Danish, Norwegian, and Swedish.

Lundström continues her analysis of 'the migrant' by discussing the concept of 'stranger danger', where danger is posited as coming from the outside, from outsiders, from what is from elsewhere. That such dynamics of 'otherness' are present in Nordic discourse is confirmed by Krzyżanowskia and Ekström, in their study of Swedish press discourse on immigration in the period 2010–2022:

> We contend that during the often closely entangled mainstreaming and normalisation processes, anti-immigration sentiments often become ever more prevalent in the public domain while the portrayal of immigration and multiculturalism as "challenges", or even as "threats" to society, is made into the acceptable "new normal" (...) The above, we claim, builds on the ever more evident recurrence of discourses wherein immigrants are framed as a default "other" of (however imaginary) native "people" and "society" (Krzyżanowskia and Ekström 2024, pp. 5–6).

Presenting something foreign as domestic, as translation is able to do, is not a neutral activity, but one that has great potential to impact and even define the dynamics around what counts as coming from the outside, and, therefore, represents danger and challenge, and what is seen as 'native' and/or rightfully dominant. As the analysis above has shown, the Bible in its translated form plays a role in the creation of imaginary native peoples and societies and of an associated idea of belonging in the Nordic context.

Venuti's work helps to make the political and ethical tensions visible by which Bible translation is done, as well as the tensions around what Bible translation does. While there is no easy solution to the tensions of translation, it is vital to recognize the consequences that translation practices are likely to have. In the case of domesticating trans-

lation, Venuti describes these as maintaining 'the status quo, reaffirming linguistic standards, literary canons, and authoritative interpretations, fostering among readers who esteem such resources and ideologies a cultural narcissism that is sheer self-satisfaction' (Venuti 1995, p. xiv). In the context of migration discourse, such tendencies are likely to contribute to discursive polarization that has damaging consequences for those who are perceived as 'other'.

Calling the Bible a migrant as I have done here might be vulnerable to similar critique as that levied against Venuti's concept of foreignization: that it is an exoticizing move which risks confirming the type of boundaries that it aims to challenge. It is not a move that I have made previously or that I am likely to make in other contexts. I use it here to create a hopefully evocative image, that domesticating translation turns the Bible into a 'successful migrant', who can have the effect of preventing others who cannot similarly perform native status, from being part or coming in.

**Funding:** This research received no external funding.

**Institutional Review Board Statement:** Not applicable.

**Informed Consent Statement:** Not applicable.

**Data Availability Statement:** Data is contained within the article.

**Acknowledgments:** I am grateful to the organizers of the Transgressing Boundaries meeting, Alexiana Dawn Fry and Ida Hartmann for their wonderful work, and to all participants for stimulating papers and conversations.

**Conflicts of Interest:** The author declares no conflict of interest.

# Notes

1.   I use the broader term 'Nordic' to avoid the suggestion that the issues considered here should be seen as specifically 'Scandinavian'. They rather likely extend beyond the three countries that are the focus in this article.

2.   Many other countries could of course be mentioned here. Kenya and the Philippines and rank first and second globally when it comes to Christian nationalism in a recent PEW study, 'Comparing Levels of Religious Nationalism Around the World' (Pew Research Center 2025). When asked specifically about the Bible, the same study shows that in Kenya, 68% percent of respondents said the Bible should have 'a great deal of influence' on the laws of their country. These numbers were 57% and 51% for Colombia and the Philippines respectively. For Sweden, the only Nordic country included in the study, the corresponding number was 8%, with 67% saying that the Bible should have 'no influence at all'.

3.   Additional relevant aspects would be the use of the Bible in Nordic media (see, e.g., Liljefors 2022; Strømmen 2024), as well as the role of children's Bibles (see, e.g., Bylund 2023).

4.   These examples are taken from previous research that provides a more comprehensive argument. See Neutel and Bjelland Kartzow (2023). National Bible societies are not the only producers of Bibles in Nordic countries, but they are the ones who make the types of claims that are central to this article. Other Bible producers tend to be more closely affiliated with specific religious communities and to have fewer national aspirations.

5.   For the complex political origin of this translation and its impact on the translation, see Pleijel (2018).

6.   On the intersection of language and politics in relation to Nordic indigenous communities, see Josefsen and Skogerbø (2021).

7.   The edited volume on the Nordic Bible by Marianne Bjelland Kartzow, Kasper Bro Larsen, and Outi Lehtipuu (Bjelland Kartzow et al. 2023) has added significantly to this scholarship.

8.   The Youtube video (https://www.youtube.com/watch?v=ZdvtCsb81Tw (accessed on 5 March 2025)); Norwegian Bible marathon (https://kommunikasjon.ntb.no/pressemelding/10971834/kronprinsesse-mette-marit-tar-aktivt-del-ved-apningen-av-bibelfestivalen?publisherId=89452 (accessed on 5 March 2025)); (https://www.bibelselskabet.dk/billedserie-kendte-og-kongelige-til-bibelselskabets-200-ars-jubilaeum (accessed on 5 March 2025)).

9.   The Bible was presented by Moderator of the General Assembly of The Church of Scotland with the words: 'Sir: to keep you ever mindful of the law and the Gospel of God as the Rule for the whole life and government of Christian Princes, receive this Book, the most valuable thing that this world affords. Here is wisdom; this is the royal law; these are the lively Oracles of God'. This and the notes to the coronation liturgy are cited on the website of the BFBS, https://www.biblesociety.org.uk/latest/news/all-the-bible-verses-in-the-coronation (accessed on 5 March 2025).

10    E.g., Burke et al. (2013); Bragg (2012). For an overview of English Bible translations apart from the King James tradition, see Naudé (2021).

11    *The God Bless the USA Bible* can be found here: https://godblesstheusabible.com/?srsltid=AfmBOorhjqwbFSY2SCXFeLyTjekMMqEKM_VlEuE6BH8vW8mlonZg0BKU (accessed on 5 March 2025).

12    Venuti does refer to historical Bible translation occasionally, specifically to the King James Bible and the German translation of the Hebrew Bible by Martin Buber and Franz Rosenzweig when discussing retranslation (Venuti 2004, pp. 26, 29).

13    Mikael Winninge, the director of translation for the Swedish NT 2026 does so in a video interview 'Varför en ny bibelöversättning' https://www.bibelsällskapet.se/forelasningar/ (accessed on 5 March 2025)), and in the introduction to *Här börjar evangeliet: pilotöversättning av Markusevangeliet, Filipperbrevet och Första Johannesbrevet*, which includes the intial results of the new translation project. Økland (2024).

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
