# Peer review of "The Bible as a Successful Migrant? Translation, Domestication, and Nordic National Identity"

_religions, doi:10.3390/rel16050647_

Round 1

Reviewer 1 Report

Comments and Suggestions for Authors

The subject and purpose of this ms was clear from the outset and I get a clean idea of its outline from the outset.  The process of presentation made sense as I moved into each section.  I note minor errors of grammar/spelling/punctuation (lines 69, 143, 174, 213) that I took as random mistakes only. Overall quality of English is impeccable. 

Author Response

Comment 1: I note minor errors of grammar/spelling/punctuation

These have been corrected

Reviewer 2 Report

Comments and Suggestions for Authors

The article is interesting and generally well-written. Its approach is original. The application of Lawrence Venuti’s theory of domestication/foreignization on the material works well. The article would be a valuable contribution to the special issue. In what follows I point to some areas of potential improvement.

The Nordic Bible Societies are accorded a primary place as Bible producers. It is noted that for Sweden the case is a bit differently due to the Swedish government’s involvment in the production of the latest official translation. In the latest decades, however, the national Bible societies have been key players in the production of vernacular bibles in all three Nordic countries. This, however, has not only entailed bibles translated into Danish, Swedish, and Norwegian. At least for Norway and Sweden, Bible Societies have also produced bibles in minority languages, for example the different Sami language varieties (which is indeed mentioned by the author on p. 5, noting that the Swedish royals were present when the North Sami translation was presented in 2020). One might also that mention the Bible is being ‘translated’ into sign language by the Swedish Bible Society. In this sense, the ‘us’ that is (implicitly or explicitly) constructed by the Bible Societies does indeed include linugistic and ethnic minorities (and here the case is probably quite different from the 1960s and 1970s when for example the Swedish translation was initiated). This fact undermines what I understand as the author’s general argument, namely that the Bible is a part of national(ist) discourse and therefore a component of social exclusion. In one way or the other, this must be addressed by the author.

Although the notion of the Bible as a “migrant” is intriguing and thought-provoking, it is not really expanded much upon. Its analytical value would have been greater had this concept been more developed (and in this way it would also have been of more value for future scholarship). I suggest that the author should at least attempt a more proper definition or some sort of conceptual discussion. The beginning of the article could preferably feature a section where the notion of the Bible as “migrant” is (tentatively) outlined.

It is also not entirely clear what the relationship between the Bible as a “migrant” and real-life migrants (i.e., people) is, although some sort of analogy seems to be suggested or at least implied. In the concluding discussion, the author quotes Lundström & Krzyzanowskia and Ekström at some length in order to suggest a connection between national(ist) Bible discourses and non-Nordic people (migrants) being constructed as outsiders, specifically as a product of translation practices. Is it the case that the Bible has been domesticated and its foreigness thereby supressed, in analogy with how real-life migrants are treated (or at least “constructed”) in contemporary Nordic societies? But if migrants are indeed “others” of Nordic national identities, would not this be the very opposite of how the Bible is being domesticated, since its foreigness is erased and it is wholly integrated in Nordic societes (hence not being constructed as “other”)?

Again, a problem for the author’s argument is the contemporary production of Nordic bibles into other linguistic forms than the majoritarian languages, which in my understanding would mean that the Bible seizes to be a (discursive) instrument for exclusion and boundary-making, instead starting to contribute to linguistic and ethnic diversity. In other words, it is not apparent that the Bible in a contemporary Nordic setting is “used to keep others out,” and if so how. If the author wants to maintain that the Bible is indeed made to function as an instrument for othering people who are not considered part of national identities of the Nordic countries, this needs to be more carefully argued and substantiated with empirical evidence.

The article rests to a certain extent on secondary references, not least from the edited volume The Nordic Bible (2023). The article would have benefitted from a more solid empirical base, where for example discourse of the Nordic Bible Societies could have been investigated more systematically. The author should at least consider adding more primary material.

Finally, it is a bit unclear how the article is positioned in relation to earlier research. Is this an article within biblical reception? If so, the number of references mentioned in footnote 4 should be expanded. If on the other hand this is an article in translation studies, then the work of Venuti could preferably be more explicitly framed in relation to this field. This could also include previous research on bible translation.

Author Response

The reviewer points out some areas of potential improvement, most of which have been taken up.

Comment 1: The Nordic Bible Societies are accorded a primary place as Bible producers. It is noted that for Sweden the case is a bit differently due to the Swedish government’s involvment in the production of the latest official translation. In the latest decades, however, the national Bible societies have been key players in the production of vernacular bibles in all three Nordic countries. This, however, has not only entailed bibles translated into Danish, Swedish, and Norwegian. At least for Norway and Sweden, Bible Societies have also produced bibles in minority languages, for example the different Sami language varieties (which is indeed mentioned by the author on p. 5, noting that the Swedish royals were present when the North Sami translation was presented in 2020). One might also that mention the Bible is being ‘translated’ into sign language by the Swedish Bible Society. In this sense, the ‘us’ that is (implicitly or explicitly) constructed by the Bible Societies does indeed include linugistic and ethnic minorities (and here the case is probably quite different from the 1960s and 1970s when for example the Swedish translation was initiated). This fact undermines what I understand as the author’s general argument, namely that the Bible is a part of national(ist) discourse and therefore a component of social exclusion. In one way or the other, this must be addressed by the author.

Response 1: This issue is discussed in a new paragraph on page 3. Unlike the reviewer, I actually see the fact that the Bible Soceties produce several different translation and yet present the ones in the majority languages with the most emphatic appeal to national significance as support for the main point of the article, rather than a challenge.

Comment 2: Although the notion of the Bible as a “migrant” is intriguing and thought-provoking, it is not really expanded much upon. Its analytical value would have been greater had this concept been more developed (and in this way it would also have been of more value for future scholarship). I suggest that the author should at least attempt a more proper definition or some sort of conceptual discussion. The beginning of the article could preferably feature a section where the notion of the Bible as “migrant” is (tentatively) outlined.

Response 2: The concept is discussed more fully in the revised version both in the introduction and the conclusion.  

Comment 3: It is also not entirely clear what the relationship between the Bible as a “migrant” and real-life migrants (i.e., people) is, although some sort of analogy seems to be suggested or at least implied. In the concluding discussion, the author quotes Lundström & Krzyzanowskia and Ekström at some length in order to suggest a connection between national(ist) Bible discourses and non-Nordic people (migrants) being constructed as outsiders, specifically as a product of translation practices. Is it the case that the Bible has been domesticated and its foreigness thereby supressed, in analogy with how real-life migrants are treated (or at least “constructed”) in contemporary Nordic societies? But if migrants are indeed “others” of Nordic national identities, would not this be the very opposite of how the Bible is being domesticated, since its foreigness is erased and it is wholly integrated in Nordic societes (hence not being constructed as “other”)?

Response 3: This is also now clarified further in the conclusion, by discussing how political exclusionary rhetoric that uses the Bible draws on the wider cultural conception of the Bible as ‘ours’ and not ‘other’. The parallel with real migrants is that some are labelled as ‘other’ while some are not based on perceived cultural belonging.

Comment 4: Again, a problem for the author’s argument is the contemporary production of Nordic bibles into other linguistic forms than the majoritarian languages, which in my understanding would mean that the Bible seizes to be a (discursive) instrument for exclusion and boundary-making, instead starting to contribute to linguistic and ethnic diversity. In other words, it is not apparent that the Bible in a contemporary Nordic setting is “used to keep others out,” and if so how. If the author wants to maintain that the Bible is indeed made to function as an instrument for othering people who are not considered part of national identities of the Nordic countries, this needs to be more carefully argued and substantiated with empirical evidence.

Response 4: This is addressed in the revisions described under response 1 and 3. The Bible is used to keep others out explicitly in political rhetoric (as clarified on page 13-14) which relies on a broader cultural sense.  

Comment 5: The article rests to a certain extent on secondary references, not least from the edited volume The Nordic Bible (2023). The article would have benefitted from a more solid empirical base, where for example discourse of the Nordic Bible Societies could have been investigated more systematically. The author should at least consider adding more primary material.

Response 5: This is definitely something that can be done in subsequent research, but the present article aims to introduce a critical perspective and includes a sufficient empirical basis for this, as the reviewer also acknowledges.

Comment 6: Finally, it is a bit unclear how the article is positioned in relation to earlier research. Is this an article within biblical reception? If so, the number of references mentioned in footnote 4 should be expanded. If on the other hand this is an article in translation studies, then the work of Venuti could preferably be more explicitly framed in relation to this field. This could also include previous research on bible translation.

Response 6: The article aims to contribute to a wide academic audience interested in (Nordic) religion, politics and migration, rather than specifically to the fields of either biblical reception or translation studies. For this purpose, there seems limited relevance in adding to the references.

Reviewer 3 Report

Comments and Suggestions for Authors

This good article has some rather conspicuous gaps in it:

  • History: The Bible has a deep history in the Nordic countries from as far back as the Middle Ages, intertwined with their saga traditions (see recently Grønlie on Iceland and Norway). The national churches (carrying the Bible) played key roles in the formation of the modern nation states. All of this deserves more than just one brief paragraph hidden in the centre of the article.
  • Media: This is surely a much better measure of a foreign cultural product’s domestication than politics or royalty!
  • Examples: The discussion of domestication and foreignization is good, but would benefit greatly from many specific examples of eadh in historic and modern Nordic Bible translations
  • "migrant … who is now used to keep others out": This most interesting comment in the abstract is unfortunately not followed up on in the article.

At least 1-2 paragraphs should be added on each of the above issues. Alternatively, the title should be narrowed to e.g. "... and Nordic public life today"

English

Just a few typos and unclear expressions:

  • cannon > canon
  • One aspects > One aspect
  • which … inscribes … inscribed > to which … subscribes … subscribed to (?)
  • the primarily the > primarily the
  • like a Nordic contemporary > as a member of contemporary Nordic society

Details

"Bible production has been dependent upon those in state power from the Emperor Constantine in the fourth century, when the very first copies were produced": This statement requires referencing—were no coopies of Bible manuscripts made before Constantine? Has not Bible distribution sometimes in history rather gone against the wishes of those in power?

Section 2 focuses so much on the Bible Societies that one is left wondering if there are no other relevant actors, and if not, why not? Further, may not the branding of a new Bible translation as a national cultural product be seen as simply a marketing strategy, rather than a sign that it truly does have that status?

Royalty and politics: It may be questioned whether these parameters give the most genuine picture of a cultural product’s immigration status. What about media, secular advertising and family life?

DO36’s concept of the four corners of a trampoline is new and genuinely interesting. It deserves greater prominence in this article. The relative importance of each ‘corner’ is of theoretical importance.

Retranslation: Some reference to the literature on this would be valuable (e.g.  Chesterman; recently Boulogne, de Lang and Verheyden).

Retranslation: This section would benefit from a comment on the difference between ‘new’ translations and ‘revisions’.

Author Response

The reviewer points out some gaps.

Comment 1:

  • History: The Bible has a deep history in the Nordic countries from as far back as the Middle Ages, intertwined with their saga traditions (see recently Grønlie on Iceland and Norway). The national churches (carrying the Bible) played key roles in the formation of the modern nation states. All of this deserves more than just one brief paragraph hidden in the centre of the article.

Response 1: I recognize that this point might be unclear and have clarified on page 1 that the issue here is the framing of the role of the Bible in the Nordic context, rather than it’s actual history, which is therefore only discussed briefly.

Comment 2:

  • Media: This is surely a much better measure of a foreign cultural product’s domestication than politics or royalty!

Response 2: While I do not agree that the use of the Bible in the media is a better measure of the Bible’s domestication, I do recognize that it should be mentioned as an additional relevant aspect and have included this on page 3. The political use of the Bible in anti-migration discourse is the central impetus for the article, which is now stressed in the conclusion.

Comment 3:

  • Examples: The discussion of domestication and foreignization is good, but would benefit greatly from many specific examples of eadh in historic and modern Nordic Bible translations

Response 3: I appreciate this suggestion and definitely see it as an interesting point to further develop, but do not see room for this within the current scope and aim of the article.

Comment 4:

  • "migrant … who is now used to keep others out": This most interesting comment in the abstract is unfortunately not followed up on in the article.

At least 1-2 paragraphs should be added on each of the above issues. Alternatively, the title should be narrowed to e.g. "... and Nordic public life today"

Response 4: I agree with and appreciate this point. The concept is discussed more fully in the revised version both in the introduction and the conclusion. 

The typos and details mentioned by the reviewer have been corrected and included where possible.

Round 2

Reviewer 3 Report

Comments and Suggestions for Authors

A few new typos:

  • supplementary
  • above-mentioned
  • Philippino